# Renal Organic Anion Transporters 1 and 3 In Vitro: Gone but Not Forgotten

**DOI:** 10.3390/ijms242015419

**Published:** 2023-10-21

**Authors:** Pedro Caetano-Pinto, Simone H. Stahl

**Affiliations:** 1Department of Urology, University Medicine Greifswald, Ferdinand-Sauerbruch-Straße, 17475 Greifswald, Germany; 2CVRM Safety, Clinical Pharmacology and Safety Sciences, R&D, AstraZeneca, 310 Darwin Building, Cambridge Science Park, Milton Road, Cambridge CB4 0WG, UK; simone.stahl@astrazeneca.com

**Keywords:** drug transporter, organic anion transporters, regulation, renal physiology

## Abstract

Organic anion transporters 1 and 3 (OAT1 and OAT3) play a crucial role in kidney function by regulating the secretion of multiple renally cleared small molecules and toxic metabolic by-products. Assessing the activity of these transporters is essential for drug development purposes as they can significantly impact drug disposition and safety. OAT1 and OAT3 are amongst the most abundant drug transporters expressed in human renal proximal tubules. However, their expression is lost when cells are isolated and cultured in vitro, which is a persistent issue across all human and animal renal proximal tubule cell models, including primary cells and cell lines. Although it is well known that the overall expression of drug transporters is affected in vitro, the underlying reasons for the loss of OAT1 and OAT3 are still not fully understood. Nonetheless, research into the regulatory mechanisms of these transporters has provided insights into the molecular pathways underlying their expression and activity. In this review, we explore the regulatory mechanisms that govern the expression and activity of OAT1 and OAT3 and investigate the physiological changes that proximal tubule cells undergo and that potentially result in the loss of these transporters. A better understanding of the regulation of these transporters could aid in the development of strategies, such as introducing microfluidic conditions or epigenetic modification inhibitors, to improve their expression and activity in vitro and to create more physiologically relevant models. Consequently, this will enable more accurate assessment for drug development and safety applications.

## 1. Introduction

Renal proximal tubule epithelial cells (RPTECs) play a critical role in the overall function of the kidney and are responsible for the secretion of waste products, reabsorption of solutes and proteins, or the regulation of acid–base balance [1,2]. One of the key players in the process of secretion are uptake transporters predominantly expressed in the basolateral membrane of RPTECs, which include organic anion transporters 1 and 3 (OAT1 and 3) [3]. These transporters are part of a wide array of proteins that regulate renal secretion and are crucial to the proper function of the kidney. OAT1 and OAT3 are responsible for the secretion of a broad range of organic anions, including several metabolic waste products and drugs, from the blood into the urine [4,5]. Evaluation of the activity of these transporters and their interaction with potential new therapeutics is important in drug development, as they play a significant role in drug pharmacokinetics and pharmacodynamics [6]. However, a significant limitation in drug testing and the study of renal physiology is the loss of OAT1 and OAT3 expression in traditional in vitro models, such as cell lines. Despite extensive research into the regulatory mechanisms of these transporters, the reasons for this persistent loss of expression in vitro are not fully understood [7,8,9,10].

OATs belong to the larger SLC22—Solute Carrier 22—family of membrane transporters, which includes organic cation transporters (OCTs), organic zwitterion/cation transporters (OCTNs), and multidrug and toxin extrusion (MATE) transporters [11]. These transporters are transmembrane proteins that facilitate the translocation of their substrates across the cell membrane through diffusion [12]. OAT1 and OAT3 utilize an exchange transport mechanism to move substrates into the cytoplasm. The binding of a substrate to the extracellular-facing domain of the transporter induces a conformational change that exchanges a molecule of intracellular α-ketoglutarate (αKG) for the extracellular substrate. OATs have a wide substrate specificity [1,6] and translocate various therapeutic small molecules, including nonsteroidal anti-inflammatory drugs (NSAIDs, e.g., aspirin, ibuprofen), antibiotics (e.g., penicillin, gentamicin), antivirals (e.g., tenofovir, cidofovir), and antineoplastic agents (e.g., methotrexate, imatinib) [13,14]. OAT1 and OAT3 are also involved in the renal clearance of uremic toxins, which are endogenous metabolic byproducts that can accumulate in the systemic circulation of patients with chronic kidney disease. These toxins are known to exacerbate various cardiovascular and metabolic conditions [15].

A significant number of approved small-molecule drugs are eliminated from the body through the kidneys [16]. Among the transporters involved in renal excretion, OAT1 and OAT3 are the most abundant anion uptake transporters expressed in the human kidney, and their broad substrate specificity has significant implications for drug development [17]. Nowadays, animal experimentation for drug testing purposes is gradually being replaced by more complex in vitro models and comprehensive approaches to reduce the use of animals in research [18,19]. Considering that most renal in vitro models currently available do not recapitulate the activity of OAT1 and OAT3, it is important to understand the mechanisms that contribute to the loss of these transporters to develop or improve in vitro tools that more accurately reflect human renal function.

## 2. Cellular and Physiological Regulation of OAT1 and OAT3 Expression and Activity

The tight regulation of OAT1 and OAT3 expression and activity involves intricate cellular pathways, including transcriptional, post-transcriptional, and post-translational processes, which maintain and modulate transport activity in response to variation in cellular homeostasis [20]. Changes in cellular energy supply (e.g., nutrient deprivation), metabolism, or the extracellular environment (e.g., oxygen levels) can trigger complex signaling pathways leading to alterations in transporter expression and may result in the permanent loss of OAT1 and OAT3 expression in vitro after isolation from renal tissue. Several in vitro and in vivo studies using multiple cell lines and animal models, respectively, have shed light on the complex regulation of these uptake transporters (Figure 1).

### 2.1. Metabolism and Hypoxia

RPTECs are highly metabolically active and require a significant amount of energy to fulfill their secretory functions. Typically, energy production occurs through mitochondrial respiration, which generates the high levels of ATP needed to power their prolific membrane transport machinery [21]. However, despite their high mitochondrial activity, RPTECs are subjected to a low-oxygen environment due to the organization of the renal vasculature, which acts as a limiting factor for oxygen diffusion into cells [22].

Hypoxia occurs when low oxygen levels disrupt normal cellular function, triggering a compensatory response to counteract its effects. Hypoxia-inducible factor (HIF)-1 plays a central role in the cellular response to hypoxia. Under normal oxygen conditions, HIF-1α, the oxygen-sensitive subunit of HIF-1, is rapidly degraded by the proteasome. However, when oxygen levels decrease, HIF 1α is stabilized and translocates to the nucleus, where it dimerizes with HIF-1β and binds to hypoxia-responsive elements (HREs) in the promoter regions of target genes [23,24]. HIF-1 increases the expression of genes involved in erythropoiesis, angiogenesis, and glucose metabolism and has been implicated in the direct and indirect regulation of OAT1 and OAT3 expression [25]. Cells can shift towards glycolysis as an alternative metabolic pathway when oxygen levels are low since this process can produce ATP in its absence. This leads to an increase in glucose uptake and consumption, which is mediated by HIF-1 and the upregulation of glucose transporters such as GLUT1 and GLUT3 into the cell [26]. The intracellular glucose concentration has been shown to affect the expression of OAT3 in the kidney. High glucose levels downregulate OAT3 expression through the activation of the transcription factor nuclear factor-kappa B (NF-κB) [27,28]. On the other hand, low glucose levels can upregulate OAT3 expression by activating the AMP-activated protein kinase (AMPK) pathway, which is a key regulator of cellular energy homeostasis [29]. Insulin-like growth factor 1 (IGF-1) seemingly plays a role in OAT3 upregulation by stimulating the activity of protein kinase A (PKA) dependent pathways, an effect that is counteracted by the pharmacological inhibition of the IGF-1 receptor [29]. The activity of this receptor may be behind the apparent relation between OAT3 deregulation and elevated glucose levels, considering that it promotes glucose uptake [30]. Ischemia-reperfusion injury, which leads to a hypoxic event in the renal proximal tubules, is also shown to decrease the activity and expression of OAT1 and OAT3 in rats [31].

The activity of OAT1 and OAT3 transporters is influenced by the intracellular concentration of αKG, which serves as a co-substrate for these transporters. αKG is synthesized within the mitochondria as an intermediate product of the tricarboxylic acid (TCA) cycle during the production of ATP [32]. Hypoxic conditions that favor glycolytic activity to the detriment of mitochondrial respiration lead to a reduction of αKG production and incidentally a reduction in OAT1 and OAT3 activity. This indirect functional regulation underlines the multiple regulatory effects that oxygen levels and hypoxia exert on the expression and activity of OAT transporters. Interestingly, αKG levels indirectly stabilize the expression of HIF 1. αKG is a precursor of succinate, which can inhibit the activity of prolyl hydroxylases (PHD) [33]. Under normal physiological oxygen levels, PHD hydroxylates HIF 1α, leading to its proteasomal degradation [22]. This mechanism keeps HIF-1 activity in check and, arguably, elevated intracellular αKG levels can enable cells to maintain HIF-1-mediated activities when oxygen levels are elevated. An important consideration is the lower oxygen pressure that the kidneys experience relative to most organs, due to their vascular architecture, therefore, the activity of the PHD-HIF axis in renal cells in response to varying oxygen levels is believed to be more resilient to hypoxic conditions [22].

### 2.2. Inflammatory and Growth Factors

Inflammation is a complex biological response to injury or cellular stress characterized by the activation of immune cells and the release of various inflammatory mediators, including cytokines, chemokines, and reactive oxygen species. Hypoxia and metabolic impairment are major drivers of inflammation [34]. The expression of OAT1 and OAT3 is also impacted by inflammation, and several immune factors activate pathways that will ultimately modulate the activity of these transporters.

The NF-κB pathway is a major mechanism by which inflammation regulates the expression of OAT1 and OAT3. This pathway is activated in response to a variety of inflammatory stimuli, including cytokines and oxidative stress. NF-κB regulates transporter expression by directly binding to their gene promoters and inducing transcription [4]. Inflammatory cytokines, such as tumor necrosis factor-alpha (TNF-α) and interleukin-1 beta (IL-1β), are potent activators of the NF-κB pathway and have been shown to upregulate the expression of OAT1 and OAT3 in renal cells [35]. The mitogen-activated protein kinase (MAPK) pathway is activated in response to cytokine release, and its activation leads to the phosphorylation of transcription factors, which in turn regulate the expression of target genes. Several MAPKs, including extracellular signal-regulated kinase (ERK), c-Jun N-terminal kinase (JNK), and p38 MAPK, have been implicated in the regulation of OAT1 and OAT3 expression [9,36]. The role of MAPK in organic anion transport regulation was further substantiated by evidence that treatment with the MAPK inhibitor U0126 lowers the expression of these transporters in renal proximal tubular cells [37].

Growth factors are signaling molecules that play important roles in the regulation of cell growth, differentiation, and survival. Recent studies have shown that growth factors can also regulate the expression of OAT1 and OAT3, acting via regulatory pathways similar to those of immune factors. IGF-1 can increase the expression of OAT1 and OAT3 in renal proximal tubular cells by activating the phosphatidylinositol 3-kinase (PI3K) pathway. Upon PI3K activation, downstream signaling molecules such as alpha serine/threonine-protein kinase (AKT) and the mammalian target of rapamycin (mTOR) are triggered, subsequently inducing the expression of the transporters [29]. Hepatocyte growth factor (HGF) is produced by mesenchymal cells and plays a role in tissue regeneration and repair. In vitro studies have demonstrated that HGF treatment can upregulate the expression of renal proximal tubular cell transporters OAT1 and OAT3. This is achieved through the activation of the c-Met receptor and downstream signaling molecules, such as PI3K and AKT [38]. Epidermal growth factor (EGF) is a crucial regulator in epithelial cells and has been shown to increase the expression of OAT1 and OAT3 in renal proximal tubular cells. In immortalized RPTECs overexpressing OAT1, the gene and protein expression of the transporter was regulated by EGFR activity via the extracellular signal-regulated kinase (ERK) pathway, showing the involvement of this pathway in the post-transcriptional regulation of organic anion activity [39].

### 2.3. MicroRNAs

MicroRNAs (miRs) are short non-coding RNA molecules that regulate gene expression by binding to target mRNAs, effectively suppressing protein translation [40]. HIF-1 has the potential to upregulate the expression of specific miRs that participate in mitigating the detrimental effects of oxygen deprivation. The expression of miR-21 is associated with HIF-1 activity in vitro [41]. Indirect evidence in mice shows that the suppression of miR-21 normalizes organic anion transport activity and facilitates the secretion of uremic toxins such as indoxyl sulfate (IS) [42].

The kidneys can sense and respond to elevated levels of metabolites produced by the gut microbiome via epidermal growth factors receptors (EGFR) and downstream signaling. The uremic toxin indoxyl sulfate (IS) is a protein metabolism by-product derived from the microbiome and an endogenous OAT1 substrate. IS indirectly activates the transcriptional activity of the Aryl hydrocarbon receptor and aryl hydrocarbon receptor nuclear translocator (AhR/ARNT) by binding to and stimulating EGFR. The activity of AhR/ARNT promotes the upregulation of OAT1 expression. The expression of miR-223 supports this apparent and complex gut-kidney communication axis by stabilizing ARNT expression and ensuring the transcription of OAT1 [43]. This mechanism induces renal secretion in response to elevated IS and maintains homeostasis by removing this potentially detrimental metabolite [43]. MiR regulation is highly species- and tissue-specific, and despite evidence of miR involvement, the regulation of OATs by miRs is still poorly understood.

### 2.4. Epigenetic Modifications

Epigenetic modifications, including DNA methylation and histone deacetylation, are important mechanisms that regulate gene expression by altering the chromatin structure and accessibility of DNA to transcription factors [44]. In recent years, studies have shown that epigenetic modifications also play a role in the expression of OAT1 and OAT3 [45,46].

DNA methylation is a process by which a methyl group is added to cytosine residues in the CpG dinucleotides of DNA. This modification is catalyzed by DNA methyltransferases (DNMTs) and can lead to the repression of gene expression. Methylation acts as a regulator of tissue-specific transactivation, where differential rates of transcription dictate the expression profiles of genes across cell types and tissues. The promoter regions of OAT1 and OAT3 contain CpG islands, which are regions of DNA with a high density of CpG dinucleotides. Ex vivo studies have shown that DNA methylation of the promoter regions of OAT1 and OAT3 can lead to their repression and decrease transport activity [47]. Interestingly, certain SLC transporters in the kidney cortex, including OAT1 and OAT3, were found hypomethylated, making the promoter regions of the genes more accessible to transcription factors [48].

Histone deacetylation is a process by which the acetyl groups on histone proteins are removed, leading to the condensation of chromatin structure and the repression of gene expression. Histone deacetylation is catalyzed by histone deacetylases (HDACs). Studies have shown that inhibition of HDAC activity can lead to the upregulation of OAT1 and OAT3 expression in renal proximal tubular cells [9,49]. The hepatocyte nuclear factors 1-alpha and 4-alpha (HNF 1α and HNF 4α) are involved in the regulation of several genes during the maturation of kidney function. This mechanism involves the recruitment of elements of the p300-CBP coactivator family (histone acetyltransferase p300/cyclic adenosine monophosphate response element binding (CREB) protein), which promotes histone acetylation and the opening of chromatin structure, leading to increased transcription of the transporters [50,51]. Despite emerging evidence, our understanding of the role of epigenetic modifications in the expression of anion transporters is still limited [52].

### 2.5. Cellular Adhesion

Cellular adhesion, which is governed by the binding of cells to each other or extracellular matrix components, is an important process in many biological functions, including cell migration, tissue development, and immune response. It plays a major role in the polarity of epithelial cells, and it has been associated with the expression of organic anion transporters OAT1 and OAT3 [53,54,55]. Evidence from 3D renal models and the reconstruction of proximal renal tubules have revealed that surface topography and matrix chemistry are critical for proper cellular differentiation and functionality [56]. The introduction of microenvironmental curvature, mimicking the tubular morphology of the nephron, improves renal function and increases the expression of drug transporters, including OAT1 [57]. Anisotropic extracellular matrix architecture promotes the structural arrangement of F-actin, reinforcing epithelial cellular morphology and increasing the expression of kidney transporters [58,59]. Studies involving the generation of kidney organoids or decellularized tissues have shown that matrix rigidity is another determinant factor in the maturation of renal cells. The appropriate matrix topography and stiffness, depending on cell type (e.g., primary cells, stem cells), will dramatically influence adhesion and the development of strong phenotypical features [60,61,62].

Caveolins (Cavs) are integral membrane proteins that are mainly described to regulate receptor-independent endocytosis. Additionally, Cavs are also reported to stabilize focal adhesions, which act as anchoring points between cells and their extracellular environment. These proteins also induce the curvature at the plasma membrane when oligomerized and interact with multiple signaling elements [63,64]. Cav-1 and Cav-3 have been implicated in the upregulation of OAT1 and OAT3, respectively [65]. This evidence shows that regulators of cell membrane adhesion and topography influence the activity of these anion transporters, potentially affecting their cellular localization [54].

### 2.6. Post-Translational Regulation and Trafficking

Post-translational regulation of OAT1 and OAT3 transporters involves a range of mechanisms that modify the proteins after they are synthesized. These modifications can impact transporter activity, localization, and stability and ultimately influence their function [66]. Phosphorylation, which involves the addition of a phosphate group to specific amino acid residues on the transporter protein, can alter the conformation of the transporter, affecting its substrate binding ability and transport activity. OAT1 activity is modulated by phosphorylation at specific sites, by protein kinase A (PKA), and by protein kinase C (PKC), which can enhance or inhibit OAT1 activity [9,27,67].

Glycosylation involves the addition of sugar molecules to specific sites on the transporter protein and it has been shown to have a significant impact on OAT1 and OAT3 function [54,68]. OAT3 has been found to undergo glycosylation at asparagine residues (N-glycosylation) and serine and threonine residues (O-glycosylation) via the activity of glycosyltransferases (GTs). O-glycosylation of OAT3 has been shown to increase the transport activity of the transporter, while N-glycosylation decreases substrate uptake. OAT1 has also been shown to undergo glycosylation, although less is known about the specific impact of this modification on its function [66].

Small ubiquitin-like modifiers (SUMOs) are a family of proteins that can covalently bind to specific lysine residues in target proteins and act as functional regulators governing, among other mechanisms, transcription and subcellular localization. This process is known as SUMOylation and is initiated when a series of ligases cooperate to attach SUMO to its target. It can be reversed by the activity of proteases that cleave the modifiers from SUMOylated proteins [69]. The activity of OAT3 is upregulated by SUMOylation, where the shuttling rate of the transporter from intracellular compartments to the plasma membrane is enhanced and, consequently, its membrane expression is increased. This upregulation is compounded by a reduced rate of OAT3 protein degradation. This functional gain also correlates with the stimulation of IGF-1 mediated PKA activity, indicating that SUMOylation is the downstream post-translational regulatory process behind OAT3 activation by this pathway [70]. SUMOylation differs from ubiquitination to the extent that tagged proteins are not degraded. The ubiquitination process adds small ubiquitin proteins to targets promoting their proteasomal degradation. It is an essential mechanism for protein turnover. Like SUMOylation, ubiquitination is described to play a role in OAT regulation [66]. PKC phosphorylates the ubiquitin ligase Nedd4-2 and directly affects the ubiquitination status of OAT1 and OAT3 with a negative impact on the stability and expression of both transporters [71,72]. Evidence from studies involving the selective pharmacological inhibition of proteasomal activity indicates that OAT3 ubiquitination is increased and associated with enhanced membrane expression and anion transport activity [73,74]. Taken together, these findings highlight an important role for both SUMOylation and ubiquitination as the actuators of non-transcriptional regulatory mechanisms of anion transport activity.

## 3. OAT1 and OAT3 Expression in Kidney Cancer

Kidney cancer is a broad term that encompasses various types of malignancies arising from different renal cells. Renal cell carcinoma (RCC) is the most common type of kidney cancer and specifically develops from RPTECs [22]. RCCs are highly inflammatory, glycolytic, and angiogenic cancers. A hallmark of RPTEC transition into RCC cells is the loss of the Von Hippel Lindau (VHL) protein, a ligase that mediates HIF degradation. In its initial stages, RCC depends on HIF-1 regulation to promote a metabolic shift toward glycolysis [75]. This change in metabolism supports the genesis of the tumor microenvironment and the secretion of large quantities of cytokines that safeguard the tumor from the activity of the cytotoxic immune cells [76]. To compensate for inefficient energy production, RCC cells upregulate glucose uptake and recruit the renal vasculature to increase the nutrient supply from the bloodstream [77].

RCC cells re-write the native regulatory mechanisms of RPTECs in a process heavily dependent on DNA methylation. Nonetheless, they retain several characteristics of their precursors, including a significant drug transport activity. Efflux activity is preserved in malignant cells while uptake activity is dysregulated [7]. This phenomenon is traditionally considered a drug-resistance mechanism that promotes cell survival [78]. The RCC phenotype seems to combine several regulatory elements that contribute to the downregulation of OAT1 and OAT3, including some of the different regulatory processes previously discussed. This fact is supported by evidence showing that the expression levels of OATs are significantly reduced in human RCC tissues compared to normal kidney tissues. Furthermore, the downregulation of these transporters is also associated with a poorer prognosis in RCC patients [79]. Although evidence is scarce, DNA hypermethylation is the likely culprit for the suppression of transporter expression in RCCs [80]. The loss of another renal SLC uptake transporter, the organic cation transporter 2 (OCT2), in RCC cells has been directly associated with the methylation of its promoter region. The inhibition of DNMT activity by 5-aza-2′-deoxycytidine was shown to recover OCT2 expression both in vitro and in vivo [80,81].

## 4. Considerations for the Functional Loss of OAT1 and OAT3

The intricate regulatory machinery of drug transporters governs their proper expression, localization, and function (Table 1). While OAT1 and OAT3 share regulatory elements with other carriers, specific pathways differ for individual transporters and transporter families. It is undeniable that the isolation of RPTECs from their native environment and subsequent culture in artificial conditions leads to the loss of not only OAT1 and OAT3 but also several other membrane drug transporters and carriers. Anion transporters stand out given their importance in renal secretion and also the persistent nature of their deregulation. Interestingly, the expression and activity of the main renal efflux drug transporters, which function in tandem with OAT1 and OAT3 to achieve the secretion of anionic molecules, is often preserved in vitro [7,39]. RPTECs are well described to retain the activities of P-glycoprotein (P-gp), Multidrug resistance proteins 2 and 4 (MRP2 and MRP4), and the Breast Cancer Resistance Protein (BCRP), all members of the ATP-binding cassette (ABC) transporter family. These transporters have significant substrate promiscuity and efflux compounds from the cells, such as uremic toxins and antibiotic and antiviral drugs previously incorporated via anion transporters. The activity of ABC transporters is driven by ATP and substrate abundance and plays a major role in RCC drug resistance. Arguably, RPTECs retain these efflux mechanisms in vitro given their importance for cellular homeostasis by removing metabolic waste and toxic compounds. The characterization of the molecular mechanisms behind the regulation of OAT transporters still represents an incomplete puzzle in which the pieces are derived from a patchwork of evidence from different animal models and cell lines, as well as clinical reports. Nonetheless, building on these findings, a rationale for the loss of these transporters is beginning to emerge.

Compelling evidence for the mechanism precluding OAT1 and OAT3 expression in vitro is the fact that RPTECs freshly isolated from kidney tissue retain the protein expression of the transporters when cultured in transwell configuration while their gene expression is lost [10,82,83]. This shows that proteins could remain functional but that gene expression is truncated, an indication that transcriptional, rather than translational, events dictate the loss of the transporters. After cells are further expanded in culture and passaged, the protein expression of the transporters is also lost. Epigenetic modifications, namely hypermethylation, are responsible for the loss of OCT2 expression in RCC, a close relative of OATs also belonging to the SLC22 family [80,84]. OCT2 expression can be partially recovered using methylase inhibitors in RCC cell lines and in vivo; however, OAT1 and OAT3 expression remains absent in the presence of methylase inhibitors in vitro [7]. These findings show that other factors beyond DNA methylation may play a role in the shutdown of anion transporters.

In conventional static culture, RPTECs are subjected to atmospheric oxygen levels, which are considered to be substantially higher than what the cells experience under normal physiological conditions. In culture, both primary and cell lines predominately rely on glycolysis for energy production to the detriment of mitochondrial respiration [85]. It is believed that this notable shift in metabolic activity serves as an adaptive mechanism to ensure cell survival. Conventional culture media is often supplemented with significant glucose concentrations, and under these conditions, glycolysis may prove to be an efficient ATP production alternative [86], since the intricate oxidative phosphorylation machinery in the mitochondria, which itself requires significant basal ATP to function, is downregulated [87].

This behavior is similar to that of RCC cells, which maintain high glycolytic activity even though they experience physiologically normal oxygen levels [88]. Similar to RCC, RPTECs in vitro enhance lactate secretion, which promotes the acidification of their extracellular environment [85]. The expression of OAT1 and OAT3 in both RCCs and RPTECs in vitro is seemingly lost following a metabolic shift. A study using immortalized RPTECs overexpressing either OAT1 or OAT3 determined that the presence of the transporters decreases glycolic activity, reduces extracellular lactate levels, and upregulates mitochondrial respiration. These effects were also accompanied by an increase in αKG synthesis and efflux [89]. Taken together, these findings indicate that organic cation activity in renal cells is deregulated when energy production is predominantly through glycolytic activity. αKG is a precursor of glutamate and glutamine and its abundance can dictate protein biosynthesis in fast-growing cells [90,91]. In predominately glycolytic cells αKG production is limited and the downregulation of organic anion transporter expression and activity could act as a mechanism to preserve the intracellular αKG levels in RCC cells.

A feature considered important for the expression of OAT1 and OAT3 is the membrane polarity of RPTECs. In vivo, these transporters are expressed on the basolateral side of the proximal tubules, facing the capillary network. In vitro, cellular structure is compromised, and polarity is lost, preventing drug transporters from properly localizing to the membrane. Nonetheless, in primary cultures and cell lines remnant transport activity persists, with the proteins found in intracellular vesicles and cellular boundaries [7,8]. Overexpression (OE) systems have been used to re-introduce OAT1 and OAT3 activity into renal cell lines [89]. Despite their advantages, these models overrepresent drug transport activity, which is dominated by the OE transporter, and operate outside of normal physiological parameters. The imbalance between OE uptake transporter and native efflux transporters means that their functionality is not within relevant in vivo ratios.

The use of Microphysiological systems (MPSs), which recapitulate the tubular architecture of the renal proximal tubule and enable continuous luminal flow, has been shown to recover organic anion transport expression and activity, albeit to a limited extent, in human primary RPTECs [8,92,93,94]. Undoubtedly, more physiologically relevant culture conditions can express these transporters, with the highly polarized nature of the reconstructed renal tubules playing a role in their recovery. Interestingly, the recovery of OAT1 expression was associated with an upregulation of HNF 4α expression [8], a factor implicated in its epigenetic regulation. The metabolic activity of RPTECs in the MPS has not been investigated in these studies, and it has therefore not been possible to establish a connection between this recovery and metabolism. Likewise, it is undetermined if the recovery of organic anion transport activity in MPS is a direct consequence of cellular polarization or the introduction of flow. Kidney organoids developed from induced pluripotent stem cells (iPSCs) that recreate complex renal structures were shown to also express OAT1 and OAT3 [95,96,97]. Typically, these organoids are cultured under static conditions, suggesting that the reconstructed RPTEC architecture could be more pertinent for the re-emergence of organic anion transporters. Examples of available human renal proximal tubule in vitro models and their anionic drug transport activity are described in Table 2. Despite the substantial body of evidence offering a reasonable picture of the regulation of OAT1 and OAT3, these findings lack significant human translation and validation. Most evidence is derived from cell lines and animal models, and to date, clinical evidence regarding the regulation of these transporters is still limited [5,79]. Depressed OAT1 and OAT3 activity is found in animal models of renal failure and is directly implicated in systemic accumulation of uremic toxins. However, the functional contribution of OATs to drug renal clearance in patients with renal failure is still unclear, and medication adjustments usually do not consider the activity of particular drug transporters [1,31].

## 5. Conclusions

The loss of organic anion transporters in RPTECs in vitro is initiated by significant environmental changes that occur during isolation and subsequent cell culture [10]. One major factor that appears to contribute to the downregulation of these transporters is a shift in energy production, favoring glycolysis in an environment with heightened oxygen (conventional in vitro cell culture) [101]. This shift in metabolism seems to be a critical factor in the suppression of transporter expression. The high glucose concentrations usually present in RPTEC cell culture media are necessary to nourish the cells, facilitate the high glycolytic activity, and are a likely driver of the physiological changes of RPTECs in vitro [102]. The re-arrangement of regulatory pathways under these conditions arguably sequesters transcription factors, such as HNF-1/4α, that are responsible for the epigenetic modifications to the promoter regions of OAT1 and OAT3 that seal the fate of the transporters. In RPTECs, the highly glycolytic activity is HIF-1 dependent [103]. Although the function of VHL in regulating HIF-1 in RCC cells is well-established, its expression and activity in vitro under atmospheric oxygen conditions in isolated RPTECs remains largely unexplored. Experimental strategies aimed at reversing epigenetic modifications, such as the use of methylase and deacetylase inhibitors, can be employed to mitigate the loss of proximal tubule physiology in vitro [104]. The use of an alternative energy source such as galactose, which produces ATP less efficiently than glucose, promotes cells to rely on mitochondrial respiration to fulfill their energy demands [105]. This activity is a better reflection of RPTEC metabolism and may also mitigate the loss of physiological features in vitro. Most evidence for the regulation of OAT1 and OAT3 originates from in vitro models that are, arguably, poorly representative of native human physiology, and animal studies. Therefore, there is a need to investigate relevant regulatory mechanisms in improved in vitro models (preferably human) as well as in a clinical setting. These approaches will also benefit our understanding of the physiological alterations that RPTECs undergo in vitro and help refine human-relevant models with a high translational value for pharmacokinetic, pharmacodynamic, and safety studies.

## Figures and Tables

**Figure 1 ijms-24-15419-f001:**
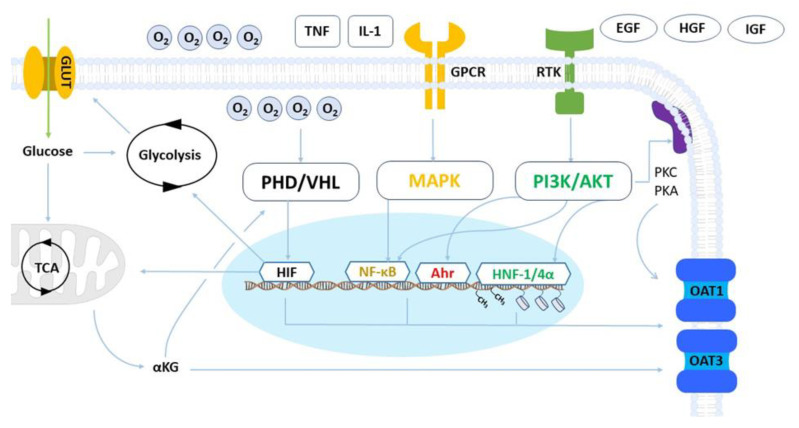
Representation of OAT1 and OAT3 regulatory networks. The prolyl hydroxylase/von Hippel Lindau (PHD/VHL) axis senses cellular oxygen levels and governs the activity of the hypoxia-inducible factors (HIF). HIF regulates the expression of genes responsible for glucose uptake via glucose transporters (GLUT) and metabolism, both glycolytic activity and mitochondrial respiration. HIF transcriptional activity can also mediate the expression of OAT1 and OAT3. Alpha-ketoglutarate (αKG) derived from mitochondrial activity is a co-substrate for OAT1 and OAT3, and its abundance influences transport activity. αKG also inhibits PHD activity and can promote HIF activation when oxygen levels are normal. G protein-coupled receptors (GPCRs) are triggered by cytokines and immune factors and act via the Mitogen-activated protein kinases (MAPKs) pathway to activate nuclear factor kappa B (NF-kB), a transcriptional factor that regulates organic anion transporter expression. Receptor tyrosine kinases (RTKs) are activated by growth factors and play a major post-transcriptional role in OAT1 and OAT3 regulation. RTKs act via the phosphatidylinositol 3-kinase/protein kinase B (PI3K/AKT) pathway to regulate gene expression via the Hepatocyte nuclear factors (HNF-1/4α), the aryl hydrocarbon receptor (AhR), and the NF-kB. RTKs also govern the activity of adaptor and adhesion proteins.

**Table 1 ijms-24-15419-t001:** Cellular pathways and molecular factors that govern the regulation of organic anion transporters and exert an up (+) or down (−)-regulatory effect.

Factors	Cellular Pathway	Process	Regulation	Effect	
				OAT1	OAT3
**αKG**	HIF-1α/VHL-PHD/TCA	Metabolism	Transcriptional	−/+	−/+
**Glucose**	HIF-1α/NF-kB/AMPK			−	−
**TNF-α**	NF-kB/MAPK	Inflammation		+	+
**IL-1β**	NF-kB/MAPK			+	+
**DNMTs**	−	Epigenetic modification		−	−
**HDACs**	−			−	−
**IGF-1**	PI3K/AKT/mTOR	Cell proliferation/ differentiation		+	+
**HGF**	c-Met/PI3K/AKT			+	+
**HNF-1α**	EGF/PI3K/AKT			+	+
**HNF-4α**	EGF/PI3K/AKT			+	+
**IS**	EGFR/AhR/ARNT			+	nd
**miR-223**	EGFR/AhR/ARNT			+	nd
**EGF**	EGFR/ERK		Post-transcriptional	+	+
**Cav**	−	Cell adhesion	Post-translational	−/+	−/+
**PKA**	SUMO	Signal transduction		−/+	−/+
**PKC**	Ubiquitin			−/+	−/+
**GTs**	−	Glycosylation		+	+

nd: not defined.

**Table 2 ijms-24-15419-t002:** Examples of in vitro human-derived renal proximal tubule models characterized for the presence (+) or absence (−) of OAT1 and OAT3 expression.

Cell Lines					Reference
Model	OAT1		OAT3		
	Expression	Activity	Expression	Activity	
Cryopreserved RPTEC	−	−	−	−	[7]
Fresh RPTEC	+	+	+	+	[10]
RPTEC-TERT1	−	−	−	−	[7]
ciPTEC	−	−	−	−	[39]
HK-2	−	−	−	−	[98]
iPSC derived RPTEC	−	−	−	−	[99]
**Advanced models**					
Nortis-ParVivo *	+	+	−	−	[8]
Emulate Kidney-chip *	+	+	−	−	[100]
iPSC derived renal organoids	+	+	+	+	[96]

* Based on cryopreserved RPTEC.

## Data Availability

Not applicable.

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
