# Peer review of "Renal Organic Anion Transporters 1 and 3 In Vitro: Gone but Not Forgotten"

_ijms, 2023, doi:10.3390/ijms242015419_

Round 1

Reviewer 1 Report

This work is carefully written review of a fascinating and not fully understood phenomena in renal transport: the loss of OAT1 and OAT3 expression in RPTEC. The review covers several physiological and pathological processes in which the loss occurs. Certainly, the review shed light on mechanisms that could help all of us because it presents the problem in an integrated picture.

Author Response

The authors are much appreciated of this positive appraisal of our manuscript

Reviewer 2 Report

This review is well organized and written. A few concerns have to be addressed.

1) The auhors should pay attention to the abbreviations,such as line  'organic anion transporters OAT1 and OAT3'.

2)  In line 204, 'decreased' should be 'decrease'. 

3) If possible, please briefly discuss the applications of OAT1 and OAT3 drugs in management of patients. 

Author Response

The authors are much appreciated of this positive appraisal of our manuscript. We have proofed our manuscript and corrected any mistakes with the use of abbreviations, as well as any minor grammatical mistakes. A brief discussion about the implications of OAT1 and OAT3 function in the choice of pharmaceutical innervations for patients with renal failure is now added in lines 418-425 of our revised manuscript

Reviewer 3 Report

Comments on the manuscript entitled “Renal Organic Anion Transporters 1 and 3 in vitro: Gone but 2 not Forgotten”

The review article by Pinto and Stahl comprehensively summarizes the current knowledge about the regulation of renal OAT1 and 3. The main issue of this article is of great relevance for the pharmaceutical research in both, academia and industry as the loss of OAT activity in frequently used in vitro model is a critical issue compromising many data sets assessed using these models.  

I have only few comments:

1.      The review is focused on OAT 1 / 3 which are responsible for the cellular uptake of several anionic compounds at the basolateral membrane of  proximal tubule epithelial cells. However, for a successful in vivo excretion, efflux transporters at the apical membrane are required. It is suggested to consider these transporters to the review because a restored OAT1/3 function alone in optimized in vitro models may be not sufficient if there is no corresponding efflux carrier.

2.      It is suggested to add a new chapter which summarizes the currently available in vitro models on renal drug transport (e.g. primary tubule cells, MDCK cells, kidney-chips) and state whether these models are affected by the mentioned loss of activity of OAT1/3. Is the compehensively described OAT regulation in these in vitro systems comparable to the in vivo situation?

3.      In addition to the mentioned aspects, what is known about the impact of transfection of other genes to the expression and localization of OAT1/3 in used in vitro models such as MDCKII cells.

4.      It remains uncertain how “a better understanding of the regulation of these transporters could aid in the development of strategies”. Here the authors need to add more precise suggestion / examples for optimized in vitro models.

Author Response

Comments and Suggestions for Authors

Comments on the manuscript entitled “Renal Organic Anion Transporters 1 and 3 in vitro: Gone but 2 not Forgotten”

The review article by Pinto and Stahl comprehensively summarizes the current knowledge about the regulation of renal OAT1 and 3. The main issue of this article is of great relevance for the pharmaceutical research in both, academia and industry as the loss of OAT activity in frequently used in vitro model is a critical issue compromising many data sets assessed using these models. 

I have only few comments:

  1. The review is focused on OAT 1 / 3 which are responsible for the cellular uptake of several anionic compounds at the basolateral membrane of proximal tubule epithelial cells. However, for a successful in vivo excretion, efflux transporters at the apical membrane are required. It is suggested to consider these transporters to the review because a restored OAT1/3 function alone in optimized in vitro models may be not sufficient if there is no corresponding efflux carrier.

Reply: We have now included a section discussing the fact that RPTEC retain the expression of relevant ABC efflux transporters that work in tandem with OAT1 and OAT3. Lines 338-348 of our revised manuscript.  

  1. It is suggested to add a new chapter which summarizes the currently available in vitro models on renal drug transport (e.g. primary tubule cells, MDCK cells, kidney-chips) and state whether these models are affected by the mentioned loss of activity of OAT1/3. Is the compehensively described OAT regulation in these in vitro systems comparable to the in vivo situation?

Reply: A new table (Table 2), lists commonly used and currently available renal and their characterized OAT1 and OAT3 expression. To the best of our knowledge, the evidence for OAT regulation is mostly derived from in vitro and animal models, with sparse studies addressing their regulation in humans. We also added a comment related to this topic to the revised version of our manuscript. Lines 416-425

  1. In addition to the mentioned aspects, what is known about the impact of transfection of other genes to the expression and localization of OAT1/3 in used in vitro models such as MDCKII cells.

Reply: A short paragraph about the implication of overexpression models is now added to the revised version of our manuscript. Lines 396-400

  1. It remains uncertain how “a better understanding of the regulation of these transporters could aid in the development of strategies”. Here the authors need to add more precise suggestion / examples for optimized in vitro models.

Reply: This sentence in the abstract is now expanded to include two precise suggestions of how to optimize OAT1 and OAT3 expression in in vitro models (microfluidic cultures and epigenetic modification inhibitors) 

Round 2

Reviewer 3 Report

The authors considered all of my questions / comments and revised their manuscipt sufficiently. I have no futher comments and recommend acceptance of the article.